# Role of Oxidative Stress and Inflammation in Doxorubicin-Induced Cardiotoxicity: A Brief Account

**DOI:** 10.3390/ijms25137477

**Published:** 2024-07-08

**Authors:** Roberta Vitale, Stefania Marzocco, Ada Popolo

**Affiliations:** Department of Pharmacy, University of Salerno, 84084 Fisciano, Italy; rvitale@unisa.it (R.V.); smarzocco@unisa.it (S.M.)

**Keywords:** cardiotoxicity, Doxorubicin, oxidative stress, inflammation

## Abstract

Cardiotoxicity is the main side effect of several chemotherapeutic drugs. Doxorubicin (Doxo) is one of the most used anthracyclines in the treatment of many tumors, but the development of acute and chronic cardiotoxicity limits its clinical usefulness. Different studies focused only on the effects of long-term Doxo administration, but recent data show that cardiomyocyte damage is an early event induced by Doxo after a single administration that can be followed by progressive functional decline, leading to overt heart failure. The knowledge of molecular mechanisms involved in the early stage of Doxo-induced cardiotoxicity is of paramount importance to treating and/or preventing it. This review aims to illustrate several mechanisms thought to underlie Doxo-induced cardiotoxicity, such as oxidative and nitrosative stress, inflammation, and mitochondrial dysfunction. Moreover, here we report data from both in vitro and in vivo studies indicating new therapeutic strategies to prevent Doxo-induced cardiotoxicity.

## 1. Introduction

Doxorubicin (Doxo) belongs to the anthracycline family, and it is a highly effective antineoplastic agent derived from the Streptomyces peucetius bacterium. Doxo is widely used in the treatment of a broad spectrum of cancers but its clinical effectiveness is hampered by cardiotoxicity, the main long-term side effect responsible for increased morbidity and mortality in cancer survivors [1]. A study conducted in the United States on women diagnosed with breast cancer showed that cardiovascular disease is the leading cause of mortality in patients over 50, exceeding the risk of death from cancer itself [2]. Therefore, patients at high risk of developing cardiac complications, including patients with hypertension, diabetes mellitus, liver disease, and previous cardiac disease, are not treated with Doxo [3]. Furthermore, Doxorubicin-induced cardiotoxicity (DIC) largely depends on the route of administration, the duration of chemotherapy (cumulative dose), and the dosage regimen [4]. Despite data on the long-term effects of different Doxo dosage regimens being quite controversial because of the different follow-up times, the recommended maximum lifetime dose of Doxo is <450 mg/m^2^ to reduce the risk of cardiotoxic side effects [5], even if some patients exhibit morphological changes (signs of cardiac damage) with a low cumulative dose of 200 mg/m^2^ [6]. An interesting systematic review summarizing 11 clinical studies concluded that a Doxo infusion duration of six hours or longer reduces the risk of clinical heart failure as well as the risk of subclinical cardiac damage compared to a faster infusion. Moreover, continuous infusion of Doxo has been correlated to less severe endomyocardial biopsy changes than those observed in patients receiving bolus infusions [7]. Reduced risk of subclinical cardiac damage has also been correlated with a Doxo peak dose of ≥60 mg/m^2^ [8]. Regarding Doxo administration schedules, Von Hoff and co-workers reported that a weekly schedule is associated with the lowest probability of developing chronic heart failure, compared with a schedule of three times per week repeated every three weeks or a schedule of once every three weeks [9]. DIC can be classified as acute or chronic. Acute DIC occurs with an incidence of about 11% and resembles acute myocarditis with myocyte damage. It usually occurs within a few days of drug administration and is regarded as being reversible [10]. On the other hand, chronic DIC manifests as cardiomyopathy and can occur months or years after the initial treatments. It is clinically characterized by an irreversible reduction in the left ejection fraction of >10% and symptomatic heart failure [1]. However, the scientific community agrees that acute and chronic cardiotoxicity represent the spectrum of a single disease and are not separate disease entities. Indeed, recent findings suggest that acute and chronic DIC are not separate events but are potentially a continuous phenomenon, starting with myocardial cell injury and followed by progressive functional decline, progressively leading to overt heart failure [11]. The time interval between treatment and the onset of chronic cardiotoxicity varies from 30 days (the so-called early onset) to more than 10 years (late onset) [10]. Therefore, heart failure in patients treated with Doxo may go unnoticed for many years. A study conducted in a cohort of 2000 cancer survivors highlighted that one-third of deaths were from long-term cardiotoxicity [12]. A recent review showed that 20% of cancer survivors in Europe show asymptomatic LV function reduction and this percentage is significantly higher in childhood survivors [13]. In asymptomatic patients, LV wall thinning, LV diameter increases, and consequent LV wall stress increases similar to dilated cardiomyopathy have been reported. Common methods used for monitoring asymptomatic cardiotoxicity include serum biological biomarkers (e.g., increased cardiac troponin and natriuretic peptide), ECG, echocardiography, and cardiac magnetic resonance. Unfortunately, these methods can only confirm that damage to the myocardium already exists [14]. In order to prevent or limit Doxo-induced cardiotoxic effects, evaluation of cardiovascular risk and active surveillance before, during, and following the therapeutic process is strongly recommended [15]. The early use of cardioprotective agents, such as β-blockers and renin-angiotensin-aldosterone system inhibitors (RAASi), including angiotensin-converting enzyme (ACE) inhibitors and angiotensin receptor blockers (ARBs), is an established practice even if data from clinical trials are not sufficient to support a clear evidence-based recommendation [16]. Therefore, knowledge of the molecular mechanisms involved and the search for therapeutic strategies able to act in the early stages of DIC are of paramount importance. Despite not yet identifying the main molecular mechanism involved in DIC, current understanding includes a central role for oxidative stress and inflammation, which are strictly related to each other and involved in different mechanisms of cell death in Doxo-treated cardiomyocytes. As described in this review, apoptosis, ferroptosis, necrosis, pyroptosis, and mitophagy are all involved in Doxo-induced cell death, and all are responsible for the loss of ventricular and later functional consequences observed in treated patients.

## 2. Involvement of Oxidative Stress in Doxorubicin-Induced Cardiotoxicity

Oxidative stress is a phenomenon caused by an imbalance between the production and accumulation of oxygen reactive species (ROS) in cells and tissues and the ability of a biological system to detoxify these reactive products. In Doxo-treated cancer patients, high levels of oxidative stress and a concomitant reduction in the antioxidant status have been observed. The nature of cardiac tissue that exhibits low levels of antioxidant enzymes, such as superoxide dismutase (SOD) and catalase (CAT), makes it more susceptible to ROS generation and the accumulation of oxidative stress [4]. Doxo-generated ROS induce cellular damage and act as a signal to activate various ways of cell death, resulting in irreversible cardiac damage. Mechanisms through which Doxo induces oxidative stress are numerous. Many studies report that the chemical structure of Doxo itself is involved in free radical generation. Doxo is well-known for its ability to produce ROS through multiple pathways, among which the redox cycle of the molecule and the formation of Doxo–iron complexes seem to play a central role. 

The redox cycle is related to the presence of the quinone moiety, which allows Doxo to act as an electron acceptor. Indeed, a one-electron reduction of the quinone moiety via NADPH and cytochrome P450 reductase converts Doxo into a semiquinone radical. The latter reacts with oxygen (O_2_), thus generating the superoxide radical O_2_^•−^ and the contemporary regeneration of the quinone form. The dismutation of O_2_^•−^ to hydrogen peroxide (H_2_O_2_) is catalyzed by SOD or can occur spontaneously. H_2_O_2_ is a relatively stable and non-toxic molecule, which is eliminated by catalase or glutathione peroxidase under physiological conditions. However, H_2_O_2_ and O_2_^•−^ can generate highly reactive and toxic OH∙ hydroxyl radicals that react with any oxidizable compound in their vicinity, causing damage to all types of macromolecules, including lipids, nucleic acids, and proteins. This occurs during the Haber–Weiss reaction, which is generally very slow unless catalyzed by transition metals, particularly iron. The Haber–Weiss reaction catalyzed by iron is called the Fenton reaction (Figure 1).
O_2_^•−^ + H_2_O_2_ ⟶ OH^−^ + OH∙ + O_2_ (Haber–Weiss reaction)
Fe^2+^ + H_2_O_2_ ⟶ Fe^3+^ + OH^−^ + OH∙ (Fenton reaction)

The second and most important pathway of free radical production involves the formation of the Doxo–Fe^3+^ complex, which can be reduced to Doxo–Fe^2+^ by various reducing systems, such as glutathione peroxidase 4 (GPx4), cysteine, etc. The Doxo–Fe^2+^ complex can react with O_2_ to generate O_2_^•−^, which, in turn, dismutes into H_2_O_2_ or undergoes the Haber–Weiss reaction, generating OH∙ [17,18]. A previous study demonstrated that the Doxo–Fe complex plays a pivotal role in DIC since it could cause lipid peroxidation through interaction with negatively charged membranes [19]. Moreover, Doxo treatment inhibits the fatty acid metabolism enzyme Acyl-CoA thioesterase 1 (Acot1) which reduces lipid peroxidation and ferroptosis by increasing GSH levels [19]. Several pieces of evidence demonstrate that the deleterious effects of Doxo on iron metabolism are also mediated by its interference with the proteins that sequester and bind cellular iron, thus leading to an increase in free iron, which could perpetuate the cycle of free radical generation [20,21] (Figure 2). Indeed, Doxo treatment induces an increase in transferrin levels, thus allowing a greater amount of iron into the cell and inactivating the iron regulatory proteins 1 and 2 (IRP1/2) [22]. Furthermore, downregulation of ABCB8, a protein responsible for iron transport outside the mitochondria, and ferritin has been demonstrated [23]. Increased free iron accumulation in mitochondria and the consequent production of lipid peroxidation and oxidative stress result in ferroptosis, which has been suggested to be the major form of regulated cell death in Doxo-induced cardiomyocyte death [24].

Doxo treatment is responsible for structural changes in the mitochondria, leading to the depletion of energy production [25,26]. The interaction between Doxo and cardiolipin, a membrane phospholipid that resides in the inner lobe of the mitochondrial membrane in cardiomyocytes, seems to be one of the main events responsible for Doxo-associated cardiotoxicity [27]. Indeed, the cationic charge of Doxo and the anionic charge of cardiolipin bind irreversibly, resulting in the accumulation of Doxo in mitochondria that leads to mitochondrial dysfunction mediated by oxidative stress [28]. The Doxo–cardiolipin complex affects the role of cardiolipin in the electron transport chain, with the consequent inhibition of several enzymes, such as cytochrome c oxidase and cytochrome c oxidoreductase. Furthermore, decreased ATP generation with consequent acceleration of oxidative phosphorylation has been reported in Doxo-treated cardiomyocytes [29,30,31]. Moreover, as previously described, ROS overproduction is associated with mitochondrial iron overload, leading to ferroptosis in cardiomyocytes [10]. Doxo also interferes with mitochondrial dynamics and the maintenance of mitochondrial integrity. It is well-known that excessive mitochondrial fission is involved in the initiation of apoptosis, while mitochondrial fusion can inhibit it [32,33]. It has been shown that upon Doxo treatment, mitochondrial fission protein 1 (Mtfp1), a GTPase also known as Mtp18 that is widely involved in mitochondrial fission, is upregulated [34]. Furthermore, it has been proven that Doxo affects mitochondrial fission by enhancing the phosphorylation of dynamin-related protein 1 (DRP1), which is a GTPase responsible for outer mitochondrial membrane scission [35,36,37]. Mitochondrial fusion, in contrast, is inhibited, since Mitofusin 1 and 2 (MFN 1/2), GTPases related to mitochondrial fusion, are downregulated during Doxo treatment [38]. Instead, Optic atrophin 1 (OPA1), another GTPase involved in mitochondrial fusion, is hyperacetylated under Doxo treatment and this effect reduces its GTPase activity and inhibits mitochondrial fusion [39]. Doxo-induced mitochondrial fission, together with oxidative stress, is associated with autophagy, a self-digestion process widely involved in the maintenance of functional mitochondria in DIC but that can exacerbate cardiotoxicity if the autophagic flux exceeds a certain threshold [40]. Indeed, it has been demonstrated that Doxo-induced mitochondrial toxicity can trigger autophagic (including mitophagic) responses in an attempt to remove damaged mitochondrial and cellular structures [41]. However, chronic Doxo treatment and/or higher doses further increase cellular damage, including the extent of oxidative stress, and may induce suppression of autophagy, resulting from exhaustion of the autophagic reserve capacity [42]. Doxo also causes a dose-dependent induction of the mitochondrial permeability transition pore (mPTP) opening, a phenomenon that releases mitochondrial Ca^2+^ and cytochrome c, contributing to apoptotic or necrotic cell death [42,43]. Indeed, an increase in the pro-apoptotic protein Bax and a concomitant reduction in antiapoptotic protein (e.g., Bcl2, caspase-9, and caspase-3) activation have been well-established in Doxo-treated cardiac cells [41]. 

Another key enzyme involved in Doxo-induced ROS production in cardiomyocytes is NADH oxidase (NOX) [44]. All NOX enzymes utilize NADPH as an electron donor and it catalyzes the transfer of electrons to molecular oxygen to generate O_2_^•−^ and/or H_2_O_2_. Under Doxo treatment, NOXs are involved in semiquinone radical formation in view of the transfer of an electron from NADH to Doxo [20,45,46]. The NOX family oxidases comprise seven members, each characterized by a specific core catalytic subunit, and all need to be associated with specific regulatory subunits for activation, while only NOX4 is constitutively active [47]. Among all NOX isoforms, NOX2 and NOX4 are the most expressed in cardiomyocytes and play a crucial role in DIC since it has been shown that Doxo promotes their expression and ROS generation [48,49]. Indeed, a previous study demonstrated that NOX2 and NOX4 genetic disruption attenuates drug-related myocardial dysfunction, reducing myocardial atrophy, cardiac apoptosis, and interstitial fibrosis [50]. Several studies demonstrated that Doxo promotes NOX2 expression and leads to phosphorylation of p47phox, a cytoplasmic subunit of NOX2. The membrane translocation of phosphorylated p47phox activates NOX2 and promotes ROS production [51]. In addition, Doxo induces the interaction between NOX4 and mitochondria, thus leading to further ROS generation, and NOX4 overexpression further induces mitochondrial division and causes NLRP3 inflammasomes-mediated pyroptosis in cardiomyocytes [52,53].

Doxo also interferes with Nuclear factor erythroid 2-related factor 2 (Nrf2), a redox-sensitive transcription factor belonging to the Cap ‘n’ collar family of transcription factors, which plays a pivotal role in the antioxidant defense system [54,55]. In resting conditions, Nrf2 is negatively regulated by Kelch-Like ECH associated protein 1 (Keap1), which binds to the Neh2 domain of Nrf2 and allows its degradation by CUL3-containing E3 ubiquitin ligase [19,55]. On the contrary, in pro-oxidant conditions, degradation of Keap1 results in the constitutive activation of Nrf2, which enters into the nucleus and, through the Neh1 domain, is able to bind antioxidant response elements (ARE), thus regulating a number of antioxidant enzymes, such as glutathione S-transferase (GST), heme oxygenase-1 (HO-1), and NAD(P)H quinone dehydrogenase 1 (NQO1) [56,57]. Several kinases, such as PKC, PI3K, and p38MAPK, are involved in Nrf2 phosphorylation and activation [58]. Other proteins involved in Nrf2 activation are p21 [59], p62 [60], Parkinson’s disease protein 7 (PARK7) [61], and Silent information regulator 1 (Sirt1) [62], which, under oxidative stress conditions, interfere with the Keap1–Nrf2 complex, thus facilitating Nrf2 activation and translocation into the nucleus. The effects of Doxo on Nrf2 are controversial. On the one hand, Doxo-induced oxidative stress results in the dissociation of the Keap1–Nrf2 complex, which causes an increase in the amount of free Nrf2 that enters the nucleus and bounds ARE, thus inducing the expression of downstream antioxidant proteins and biphasic detoxification enzymes [63]. An in vivo study revealed that Nrf2 activation reduces the Doxo-induced oxidative impact on cardiomyocytes [50]; however, this weak upregulation is not enough to offset the Doxo-induced oxidative stress [27]. On the other hand, Doxo administration can increase Keap1 levels [64], as well as the expression of tripartite motif-containing 21 (TRIM21), an E3 ubiquitin ligase that interacts with p62 to disturb the dissociation of Nrf2 from the Keap1–Nrf2 complex [65]. Overall, another study showed that Doxo can inhibit Nrf2 through p38 MAPK, leading to an increase in apoptosis [66]. Furthermore, it is important to note that excessive activation of Nrf2, as observed under Doxo treatment, increases heme to ferrous iron and carbon monoxide, as well as biliverdin, in a HO-1-catalyzed manner, thus triggering ferroptosis [67] (Figure 3).

In addition to oxidative stress, there is evidence that nitrosative stress is also involved in DIC, as increases in reactive nitrogen species, most notably peroxynitrite (ONOO^−^), a potent reactive and cytotoxic free radical, have been reported [4]. Nitric oxide (NO) is required for normal cardiac function, including coronary vasodilatation, inhibition of platelet aggregation and neutrophil adhesion and activation, and modulation of cardiac contractile function. In oxidative stress conditions, the inactivation of cytoprotective NO and the formation of peroxynitrite (ONOO^−^), produced following the interaction between NO and O_2_^•−^, occur [68]. ONOO^-^ has a high affinity for tyrosine residues in proteins and can form nitrolated proteins by nitrating tyrosine groups [69]. A reaction between ONOO^−^ and DNA induces the production of several oxidation products of the purine and pyrimidine bases, such as 8-nitroguanine, an oxidative DNA damage marker [70]. Moreover, ONOO^−^ has the potential to transform into NO_2_^−^, NO_3_^−^, and OH^−^, with the consequent generation of active nitrogen species (RNS) ([71]). High levels of ROS/RNS induce membrane lipid peroxidation and membrane damage and trigger death cells and apoptosis [72], potentially contributing to the Doxo-induced depression of cardiac function [69]. The activity of Doxo on nitric oxide synthetase (NOS) and enzymes responsible for NO formation, starting from L-arginine and O_2_, also contributes to the increase in nitrosative stress. To date, three isoforms of NOS have been identified: neuronal (NOS1), inducible (NOS2), and endothelial (NOS3). NOS1 and NOS3 are constitutive enzymes controlled by Ca^2+^ concentrations; NOS2, on the other hand, is Ca^2+^-independent and its expression is induced at the transcriptional level by pro-inflammatory stimuli, including NF-kB, the main nuclear transcription factor involved in inflammation [68]. Treatment with Doxo causes an increase in eNOS transcription. In this regard, Neilan et al. [73] demonstrated that cardiac-specific overexpression of eNOS exacerbates the pathological response to Doxo in the heart. Doxo binds to the eNOS enzyme and induces the formation of the semiquinone radical which, in turn, reduces oxygen to the superoxide radical [74]. Specifically, Doxo binds the reductase domain of eNOS, thereby increasing superoxide production and reducing NO formation, further aggravating oxidative stress [27,75]. Although the data on the effects of Doxo treatment on iNOS activity are controversial, evidence indicates that Doxo leads to increases in iNOS protein and mRNA expression in the myocardium, which leads to the generation of NO [76]. However, in oxidative stress conditions, the NO produced reacts with •O_2_, thus generating the cytotoxic free radical peroxynitrite anion (ONOO^−^) that induces nytrotyrosine formation with increased mitochondrial superoxide levels in cardiac tissue [19,69,77,78,79,80,81]. Regarding nNOS, its role in DIC has been minimized because no changes in myocardial nNOS mRNA expression were observed upon Doxo administration [82].

## 3. Involvement of Inflammation in Doxorubicin-Induced Cardiotoxicity

A growing body of evidence indicates that inflammation also plays a pivotal role in DIC. Previous studies showed that Doxo treatment induces a significant increase in pro-inflammatory cytokines, such as tumor nuclear factor (TNF)-α, interleukin (IL)-1β, and IL-6 via nuclear factor-κB (NF-κB) activation [82,83]. NF-κB is a downstream effector of Toll-like receptor 4 (TLR4) signaling, which is part of the innate immune system and can be activated by pathogen-associated molecular patterns (PAMPs), e.g., LPS, or by damage-associated molecular patterns (DAMPs) such as endogenous high-mobility group protein box 1 (HMGB1) and the heat shock protein family (Hsps). It has been demonstrated that Doxo-induced ROS overproduction upregulates NF-κB expression, thus promoting inflammatory factor release and aggravating the inflammatory response [84]. Moreover, studies showed that Doxo treatment induces both an up-regulation of TLR4 expression and the release of PAMPs and DAMPs. HMGB1 is secreted in response to Doxo-induced ROS and RNS production, and once extracellularly released, it can activate TLR2 and TLR4, upregulate their expression, and further activate the release and secretion of inflammatory cytokines [85,86]. It is important to note that TLR4 activation is also involved in the apoptotic pathway. Indeed, a study reported that Doxo-induced oxidative stress is associated with an increase in TLR4 expression, further promoting inflammation that, in turn, contributes to apoptosis. These data support the notion that both inflammation and oxidative stress contribute to Doxo-induced cardiomyocyte damage, creating a vicious circle of impaired pathways [87,88]. Furthermore, Doxo promotes NLR family pyrin domain containing 3 (NLRP3) inflammasome formation in both a TLR4-dependent manner and in a TLR4-independent but ROS-dependent manner [25,89,90]. ROS overproduction induced by Doxo treatment activates the NLRP3 inflammasome in cardiomyocytes and macrophages. Activation of NLRP3 inflammasomes begins with the assembly of NLRP3, apoptosis-associated speck-like protein containing a caspase recruitment domain (ASC), and pro-caspase 1, thus leading to caspase-1 activation and thereby mediating the event of pyroptosis [89,90,91]. Pyroptosis is a form of pro-inflammatory cell death characterized by cytoplasmic swelling and plasma membrane rupture induced by the cleavage of gasdermin D (GSDMD) or GSDME, which allows the release of IL-1β and IL-18, contributing to cardiac cell damage [92,93,94]. Indeed, pyroptosis can usually result in increased inflammation and can cause the activation of various caspases (e.g., caspase-1, caspase-3, caspase-4, and caspase-11). A study conducted by Ye et al. [94] found that Doxo also directly binds GSDMD, leading to plasma membrane rupture even in the absence of inflammatory caspases. Indeed, Doxo-induced pyroptosis of cardiomyocytes can also occur through the mitochondrial pathway, since Doxo leads to increased expression of the Bcl-2/adenovirus E1B interacting protein 3 (Bnip3) protein in mitochondria, which in turn activates caspase-3 [25,95]. Furthermore, Doxo-induced GSDMD-N localization in mitochondria contributes to mtDNA release and cell damage [94]. It has been shown that Doxo treatment also stimulates the formation of the TLR2–MyD88 complex, which activates the NF-κB pathway. Nuclear translocation of NF-κB induces the expression of inflammatory chemokines and cytokines (including TNF-α, IL-1β, IL-6, IL-8, IL-12, and IL-17), thus leading to cardiac inflammation and fibrosis [96]. Emerging data support the role of NLRP3 and MyD88 pathways in chemotherapeutic-induced myocardial inflammation since a recent study also showed the involvement of these pathways in immune check-point inhibitor treatment in preclinical models [97] (Figure 4).

## 4. Therapeutic Strategies to Counteract Doxorubicin-Induced Cardiotoxicity

At present, there are no specific clinical practice guidelines for the management of DIC. To reduce Doxo-induced adverse effects, especially at the cardiac level, several strategies have been proposed. 

Encapsulating Doxo in nanostructures is one of the strategies applied in recent decades to decrease cardiotoxicity, owing to changes in tissue distribution and the rate of drug release. The first Doxo liposomal injection, named Doxil^®^, was approved by the FDA in 1995. Doxil^®^ consists of a PEGylated liposomal system into which Doxo is incorporated. This formulation shows clinical advantages such as an increase in the half-life of Doxo, a reduction in adverse effects, and accumulation of the drug in tumor tissue [98]. Doxil^®^ has been shown to have reduced cardiotoxic effects compared to Doxo; however, its cumulative dose remains a clinical concern [99]. Indeed, despite a randomized trial that reported that less than 2% of patients developed cardiotoxicity when the cumulative dose reached 1061 mg/m^2^, it is recommended that the Doxil^®^ dose should not exceed 550 mg/m^2^ [100,101]. However, a 10-year follow-up showed that cumulative doses of over 550 mg/m^2^ of Doxil^®^ alone or Doxil^®^ combined with previously administered Doxo did not lead to drug-related heart failure [102]. Doxil^®^ was first approved for the intravenous treatment of Kaposi’s sarcoma; subsequently, the comforting results obtained led to an extension of the approved treatment (e.g., breast and ovarian cancer) and to the development of a similar product (Lipodox^®^) [98]. In order to reduce DIC, liposomal Doxo-based chemotherapy has also been evaluated. In 2015, a meta-analysis by Xing and co-workers analyzed data from ten clinical trials evaluating the clinical efficacy and cardiotoxicity of liposomal Doxo-based chemotherapy compared to conventional Doxo in breast cancer patients. Despite some controversial data between studies, the meta-analysis came to the conclusion that liposomal Doxo-based chemotherapy showed significant advantages in progression-free survival and reduced cardiotoxicity compared to conventional Doxo [103]. Other types of Doxo nano-formulations, such as Doxo-conjugated poly-aspartic acid/polyethylene glycol micelles (NK-911) and Doxo-loaded polyalkylcyanoacrylate nanoparticles (Livatag), are currently in clinical trials [104]. In addition, it has been demonstrated that peptide-based hydrogel (HG) and nanogel (NG) formulations are convenient approaches for drug delivery as modulating the ratios of the peptide components altered Doxo release. Indeed, HGs could be used for transepithelial drug delivery or in situ gelation process implant formation, while nanogels could be used for systemic, oral, and pulmonary drug delivery [105,106]. Magnetic iron oxide nanoparticles (IONPs) stabilized with trimethoxysilylpropyl-ethylenediamine triacetic acid (EDT) has been evaluated as a Doxo carrier for Glioblastoma multiforme (GBM) [107]. It is important to note that Doxo-IONP showed increased Doxo release under acidic conditions, such as those in the tumor microenvironment [108,109]. Based on these encouraging results, several IONPs have been produced as effective nanocarriers for anti-cancer drugs such as Doxo [110], paclitaxel [111], and 5-fluorouracil [112], although none of these are yet in clinical trials. 

Regardless of the therapeutic approaches used to prevent Doxo-induced oxidative stress and inflammation, we report data from several clinical (Table 1) and preclinical studies (Table 2).

The iron chelator Dexrazoxane has been approved by the FDA to prevent anthracycline-mediated cardiotoxicity in cancer patients [113]. Indeed, Dexrazoxane is able to bind iron before it enters cardiomyocytes, thus preventing the formation of the Doxo–Fe^3+^ complex, free radical formation, and cardiac damage; however, its use is hampered by severe side effects which are very similar to the side effect profile of anthracyclines [114,115].

To prevent Doxo-induced oxidative stress, the use of some antioxidant drugs has been proposed. As previously described, Carvedilol is widely used to prevent DIC in view of its well-known effects on LVEF and serum biological markers of cardiotoxicity [16]. However, Carvedilol, unlike other β1-receptor antagonists, showed some other cardioprotective effects. Indeed, in vitro and in vivo studies showed that Carvedilol inhibits the formation of ROS and lipid peroxidation by scavenging oxygen free radicals and preventing the consumption of endogenous antioxidants through a mechanism similar to that of Vitamin E and glutathione [63,116,117]. In addition, data from clinical trials support the use of Carvedilol in DIC prevention and the authors point out that the antioxidant effect of the drug seems to play a key role in reducing cardiotoxic effects [118,119,120,121]. In this context, the cardioprotective effects of Vitamin E have also been evaluated; however, the results have been inconclusive [122]. Statins could be used as cardioprotective agents in view of their pleiotropic effects ranging from upregulation of SOD_2_ to attenuation of ROS production, as demonstrated in two separate studies of DIC in mice, which assessed the cardioprotective role of Fluvastatin and Rosuvastatin, respectively [123,124]. Another in vivo study showed that Rosuvastatin administration in Doxo-treated rats inhibited the expression of HMGB1, in addition to the decrease of TNF-α and IFN-γ and increase of IL-4 and IL-10 [125]. However, clinical trials showed controversial results. Indeed, while the STOP-CA randomized clinical trial showed that Atorvastatin reduced the incidence of cardiac dysfunction in patients with lymphoma receiving anthracycline, another clinical trial (PREVENT) showed that Atorvastatin demonstrated very modest effects on oxidative/nitrosative stress biomarkers [126,127].

**Table 1 ijms-25-07477-t001:** Main data of clinical studies conducted to evaluate the cardioprotective effects of several drugs in Doxo-treated patients.

Trial	Drug	Study Population	Results	Ref.
CarDHA trial	Carvedilol + DHA starting 2 and 7 days before chemotherapy	32 breast cancer patients (aged 18–75 years)	Improved LVEF	[120]
Carvedilol Administration Can Prevent Doxorubicin-Induced Cardiotoxicity: A Double-Blind Randomized Trial	Carvedilol daily during chemotherapy	70 breast cancer patients	Improved LVEF	[118]
Cardioprotective Effects of Carvedilol in Inhibiting Doxorubicin-induced Cardiotoxicity	Carvedilol daily starting 24 h before chemotherapy	91 breast cancer patients (aged 21–69 years)	Reduced troponin I levels	[121]
Prophylactic use of carvedilol to prevent ventricular dysfunction in patients with cancer treated with doxorubicin	Carvedilol during chemotherapy	154 cancer patients	Improved LVEF	[119]
A prospective study to evaluate the efficacy and safety of vitamin E and levocarnitine prophylaxis against doxorubicin-induced cardiotoxicity in adult breast cancer patients	Vitamin E three times daily Levocarnitine four times daily before chemotherapy	74 breast cancer patients	Reduced CKMB No significant effects on LVEF	[122]
STOP CA	Atorvastatin daily starting before Doxo	300 patients with lymphoma	Improved LVEF	[126]
PREVENT	Atorvastatin daily starting 48 h before Doxo	279 cancer patients	No changes in LVEF Modest effects on oxidative and nitrosative stress biomarkers	[127]

For some years now, many natural compounds with antioxidant activity have been evaluated as potential cardioprotective agents, but no clinical data are currently available. Lu and co-workers showed that Resveratrol in combination with FGF1 markedly increased the nuclear accumulation of NRF2 and upregulated the expression of downstream antioxidant target genes, which ameliorated Doxo-induced myocardial injury, apoptosis, inflammation, and oxidative stress in mice [128].

Furthermore, suppression of Doxo-induced ferroptosis, possibly through modulation of the MAPK signaling pathway, has been shown for Resveratrol in a mouse model of DIC [129]. A study by Abdel-Daim and co-workers demonstrated that Allicin pre-treatment significantly reduced Doxo-induced pro-inflammatory cytokines and free radical overproduction in mice [130]. Several studies reported interesting results for Ginsenoside Rh2, which showed antioxidant and anti-inflammatory effects in Doxo-treated wild-type mice [131] and in breast cancer-bearing mice [132].

In vitro and in vivo studies showed that Neferin [133], Astragaloside IV [134], Acacia hydaspica [135], and Resolvin D1 [136] all reduce DIC by inhibiting NOX activity. However, even though numerous pieces of evidence have indicated the role of NOXs in Doxo-induced oxidative stress, no clinically effective NOX inhibitors have been developed. It has been proposed that Valsartan, an Angiotensin-converting enzyme inhibitor, could be used as a cardioprotective agent in DIC since it has been shown that AngII activates and regulates NOX expression [137]. Further studies will be needed to determine whether NOX inhibitors are valuable in clinical settings [19].

Dapagliflozin downregulates oxidative stress accumulation and mitochondrial dysfunction through the restoration of antioxidant enzymes (e.g., HO-1, NQO1, and SOD) via the PI3K/Akt/Nrf2 axis [138]. An in vivo study showed that Empagliflozin reduced Doxo-induced ferroptosis, fibrosis, apoptosis, and inflammation in mice, and the interaction between Empagliflozin and the NLRP3 and MyD88 pathways has been proven [139].

The reduction of Doxo-induced ROS formation and enhancement of antioxidant enzymes (SOD, CAT, and GSH) have been reported for Ursolic acid [140]. Upregulation of the Nrf2/ARE axis and consequent reduced ROS production have also been reported for Panax Ginseng and its ginsenosides [141]. Furthermore, antiapoptotic effects have been reported for Panax Ginseng in Doxo-treated cardiomyocytes [142]. Quercetin has been demonstrated to counteract ROS production in DIC and to prevent the opening of mPTP [143]. Several in vivo studies showed that Rutin and Apigenin administration significantly reduces Doxo-induced apoptosis and autophagy [144,145]. Moreover, it has been reported that Selenium reduces Doxo-induced inflammation via enhanced Nrf2 expression, which weakens NLRP3 activation [146].

**Table 2 ijms-25-07477-t002:** Main data of preclinical studies conducted to evaluate cardioprotective effects of different molecules in Doxo-induced cardiotoxicity models.

Model	Treatment	Effects		Statistical Analysis	Ref.
Male Swiss albino mice (10 weeks of age)	Allicin (10 and 20 mg/kg) once daily for 2 weeks Doxorubicin (10 mg/kg) on the 7th, 9th, and 11th days.	⬇AST, LDH, CK, CK-MB ⬇IL-1β, TNF-α, 8-OHdG, ⬇COX-2, caspase-3	⬆GSH,CAT, SOD, GPx	One-way ANOVA followed by post-hoc Duncan’s test.	[130]
Male Sprangue Dawley rats (220–225 g)	*Acacia hydaspica* (200 and 400 mg/kg) once daily for 6 weeks Doxorubicin (3 mg/kg) single dose/week	⬇AST, LDH, CK, CK-MB	⬆POD,CAT, SOD, QR	One-way ANOVA followed by Tukey’s test.	[135]
Male C57BL/6J mice (22.5–23.5 g)	Resveratrol (20 mg/kg/day) two weeks before Doxo injection Doxorubicin (cumulative dose of 24 mg/kg)	⬇PTGS2, ACSL4, NCOA4, ⬇p-ERK/ERK, p-p30/p38, ⬇pJNK/JNK,LDH	⬆GSH, GPX4	Shapiro–Wilk test and Brown–Forsythe test. Ordinary ANOVA or Welch ANOVA test.	[129]
H9C2	Resveratrol (20 µM) 6 h before Doxorubicin Doxorubicin (1 µM) 24 h	⬇p-ERK/ERK, p-38/p-38, p-JNK/JNK, ⬇LDH,PTGS2, ACSL4, NCOA4	⬆GSH, GPX4		
Male Sprangue Dawley rats (8 weeks)	Valsartan (20 mg/kg) daily for 6 weeks Doxorubicin (2.5 mg/kg) once per week for 6 weeks	⬇mRNA expression levels of NOX1,NOX2, NOX4, Bax, Caspase-3, (MMP)2,MMP9, Collagen I, Beclin-1, BNP,ANP, β-MHC, GDF15, TPM1, BGN, POSTN. ⬇ALD, TNF-α, IL-6, BNP,ROS, MDA	⬆BCL2, Collagen III ⬆SOD	One-way or two-way ANOVA followed by Bonferroni’s post-hoc comparisons	[137]
H9C2	Valsartan (5 or 10 µM) 1 h of pre-treatment Doxorubicin (1 µM)	⬇ROS			
Male C57BL/6J mice (10–12 weeks)	Rosuvastatin (100 µg/kg) 1 week before Doxorubicin and for further 2 weeks. Doxorubicin (10 mg/kg) single dose	⬇CAT ⬇NCX1, Ryr2	⬆p-PLN	Unpaired Student *t*-test. ANOVA and the Tukey–Kramer multiple comparison post-hoc test	[125]
Sprangue Dawley rats	Dapagliflozin (0.1 mg/kg) per day Doxorubicin (3 mg/kg) four weekly	⬇p-Smad3, BNP, α-SMA, p-38, ⬇NF-kB-p65, IL-8		One-way ANOVA followed by Bonferroni’s post-hoc comparisons	[138]
H9C2	Dapagliflozin (0–20 µM) Doxorubicin (10 µM) for 24 h	⬇ROS, pSmad3/Smad3, ANP, BNP, α-SMA, Collagen I, Fibronectin, NF-kB-p65, IL-8	⬆p-AKT, HO-1, NQO1, SOD		
C57BL/6J male mice	Resveratrol (10 mg/kg)+ Fibroblast growth factor-1 (0.5 mg/kg) for 7 consecutive days Doxorubicin (20 mg/kg) single injection	⬇CK,LDH, cTnI, cleaved caspase-3 ⬇IL-1β,IL-1α, TNF-α,Mcp1, p-IKBα, p65, ROS	⬆BCL2/BAX ratio, CAT, SOD1,SOD2, HO-1,NQO1,Sirt1	One-way ANOVA followed by post-hoc pairwise comparisons using Tukey’s test.	[128]
H9C2	Resveratrol (20 µM) Doxorubicin (1 µM)		⬆HO-1, Nrf2 (effects were canceled by *Sirt1*-shRNA		
C57BL/6J male mice (8 weeks)	Rutin (100 mg/kg) for 11 weeks Doxorubicin (3 mg/kg) every other day for 2 weeks starting after one week administration of Rutin	⬇LC3 II, ATG5, P62, ⬇Caspase-3	⬆Akt, Bcl-2	One-way ANOVA	[145]
C57BL/10 mice (10 weeks)	Fluvastatin (100 mg/kg) 4 days before Doxorubicin application Doxorubicin (20 mg/kg) for 5 days	⬇TNF-α, ⬇Bax	⬆SOD2.Bcl-2	Kruskal–Wallis test in conjunction with the Mann–Whitney *U* post-hoc test	[123]
Male Swiss mice and male Sprague Dawley rats	Ginsenoside Rh2 (5 mg/kg, 10 mg/kg,20 mg/kg) total 8 doses Doxorubicin (3 mg/kg or 2 mg/kg) cumulative dose 18 mg/kg or 8 mg/kg	⬇AST,CK,LDH, MDA	⬆SOD, CAT, GSH	One-way ANOVA; Student *t*-test was performed	[131]
C57BL/6 mice (6–8 weeks)	Resolvin D1 (2.5 µg/kg) 30 min before Doxorubicin and every day thereafter for the duration of the experiment Doxorubicin (20 mg/kg) once	⬇LDH,CK-MB, cTnI ⬇IL-1β,IL-6, NF-kB ⬇MDA, NOX2, NOX4, GRP78, CHOP, caspase-12,Bax, c-caspase3	⬆SOD,GSH, Nrf-2, OH-1,Bcl-2	One-way ANOVA followed by Tukey’s test	[136]
C57BL/6J mice (8–10 weeks)	Selenium (0.2 mg/kg) 2 weeks Doxorubicin (15 mg/kg) 2 weeks	⬇CTnI, CK, LDH ⬇IL-1β,TNF-α, IL-18, NLP3,ASC, Caspase-1	⬆SOD, GSH ⬆mRNA level of Nrf-2, HO-1,NQO-1, GCLM	One-way ANOVA with Tukey’s post-hoc test	[146]
Male Wistar rats (250–300 g)	*Panax ginseng* (5 g/kg) for 30 days Doxorubicin (2.5 mg/kg) for 2 weeks	⬇MDA	⬆GSHPx, SOD	One-way ANOVA followed by Scheffe’s multiple range test	[142]
Kunming mice	Apigenin (125 or 250 mg/kg) for 17 days Doxorubicin (3 mg/kg) cumulative dose 24 mg/kg	⬇LDH, CK,AST ⬇Bax/Bcl-2 ratio ⬇Beclin1, LC3B II/I	⬆PI3K/AKT/mTOR pathway	One-way ANOVA with LSD post-hoc test	[144]
Male Sprague Dawley rats (6–8 weeks)	Rosuvastatin (1 mg/kg) for 6 weeks Doxorubicin (1 mg/kg) for 2 weeks	⬇AST, CK-MB ⬇HMGB1, RAGE ⬇TNF-α, IFN-γ	⬆IL-10, IL-4	One-way ANOVA followed by Tukey’s post-hoc test	[125]
BALB/c female mice	Ginsenoside Rh2 (20 mg/kg and 30 mg/kg) every day Doxorubicin (2 mg/kg) cumulative dose of 22 mg/kg	⬇Caspase-3,Capsase-7, caspase-9 ⬇IL-1β, TNF-α, IL-6 ⬇α-SMA, Smad2, Smad3	⬆p 21	One-way ANOVA followed by Tukey’s multiple comparison test	[132]
H9C2 HCF HUVEC	Ginsenoside Rh2 (2.5, 5 and 10 µg/mL) 7 days after Doxorubicin treatment Doxorubicin (100 nM) for 7 days	⬇α- SMA, Vimentin ⬇MMP2, MMP4, MMP9, MMP14, MMP17, MMP19,MMP23, MMP27, MMP28			
H9C2	Neferine (10 µM) Doxorubicin (1 µM)	⬇NOX2, ROS, p-ERK, p-p38 ⬇Ca^2+^ intracellular accumulation ⬇COX2, TNF-α, Cytochrome c, Bax	⬆Cyclin D1 ⬆Bcl-2	One-way ANOVA followed by Tukey’s multiple comparison test	[133]
C57Bl/6 (6 weeks)	Empagliflozin (10 mg/kg) daily for 3 days alone and then in combination with Doxorubicin Doxorubicin (2.17 mg/kg) daily for 7 days	⬇Ferroptosis, MDA ⬇IL-1β, IL-6, IL-8, MyD88, NLRP3 ⬇MMP-9, Caspase-3		Non-parametric test. ANOVA test.	[139]
HL-1	Empagliflozin (50, 100 and 500 nM) Doxorubicin (0.1 to 50 µM)	⬇intracellular Ca^2+^, MDA, 4-HNA, NO ⬇IL-1β, IL-6, IL-8 ⬇MyD88, NLRP2			

Abbreviations: CAT, catalase; CK, Creatine kinase; CK-MB, Creatine kinase-myocardial B fraction; GPx, Glutathione Peroxidase; GSH, Glutathione; LDH, Lactate dehydrogenase; IL-1β, Interleukin-1β; MDA, Malondialdehyde; SOD, superoxide dismutase; TNF, Tumor necrosis factor; 8-OHdG, 8-oxo-2′-deoxyguanosine; AST, Aspartate transaminase; POD, peroxidase; QR, Quinone reductase; PTGS2, prostaglandin endiperoxide synthase2; ACSL4, acyl-CoA synthetase long-chain family member; NCOA4, nuclear receptor coactivator 4; BNP, brain natriuretic peptide; ANP, atrial natriuretic peptide; (MMP)2, metallopeptidase; β-MHC, β myosin heavy chain; GDF15, growth differentiation factor 15; TPM1, tropomyosin1; BGN, biglycan; POSTN, periostin; ALD, aldosterone; NOX, NAD(P)H oxidase; NXC1, Na^2+^-Ca^2+^ exchange protein; p-PLN, phospholamban; α-SMA, α-smooth muscle actin; Mcp1, monocyte chemoattractant protein-1; cTnI, cardiac troponin I; Bcl-2, B cell lymphoma-2; Bax, BCL2-associated X; GRP78, Glucose-regulated protein 78; CHOP, C/EBP homologous protein; NLRP3, NLR family pyrin domain containing 3 protein (NRLP3) inflammasome; ASC, apoptosis-associated speck like protein containing a caspase recruitment domain; MMP, matrix metalloproteinases; MyD88, Myeloid differentiation factor-88. ⬇ denotes reduce and ⬆ denotes increase.

## 5. Conclusions

Cardiotoxicity remains the main long-term side effect of many chemotherapeutic drugs, especially Doxo. Indeed, Doxo has detrimental effects, classified as acute and chronic abnormalities, including arrhythmias, heart failure, and ventricular dysfunction. In the last decade, much attention has been paid to the early events involved in cardiomyocyte toxicity, since it has been well-established that these are the first steps of a continuous escalation that degenerates into heart failure over the years. A growing body of evidence indicates that oxidative stress and inflammation are the main players involved in DIC. However, they are closely related to each other, and the activated pathways converge at many time points, thus making it complicated to establish a cause–effect relationship and identify the target on which to act. The knowledge of molecular mechanisms involved in the early stage of DIC could help to identify drugs or natural products capable of counteracting these effects, reducing the onset of chronic cardiotoxicity and cardiac dysfunction in Doxo-treated patients. As described in this review, although many drugs are used to treat overt DIC, the only FDA-approved cardioprotective agent is Dexrazoxane; however, even this is not free from side effects. Recently, several Doxo nanoformulations demonstrated better anticancer activity compared to the free drug; however, there are no definitive data on their long-term cardiac safety. A better understanding of the molecular mechanisms involved in cardiotoxicity would allow for the use of drugs currently in use for other therapeutic indications, such as Carvedilol or Statins, which have demonstrated interesting antioxidant and anti-inflammatory activity in several preclinical studies. Data from clinical trials confirm the prophylactic use of Carvedilol to reduce DIC; however, the main gap of these results is the heterogeneity of patients enrolled and the different follow-ups. The data for statins also appear to be encouraging, although there are differences between data from preclinical studies and those from clinical trials, which may be due to the different molecules used. Natural compounds with antioxidant activity seem to be a promising source, but numerous studies are needed to identify compounds that can then be evaluated in humans. However, it is now clear that cardiotoxicity is also associated with other chemotherapeutic drugs; therefore, research in this area has a large impact on clinical practice.

## Figures and Tables

**Figure 1 ijms-25-07477-f001:**
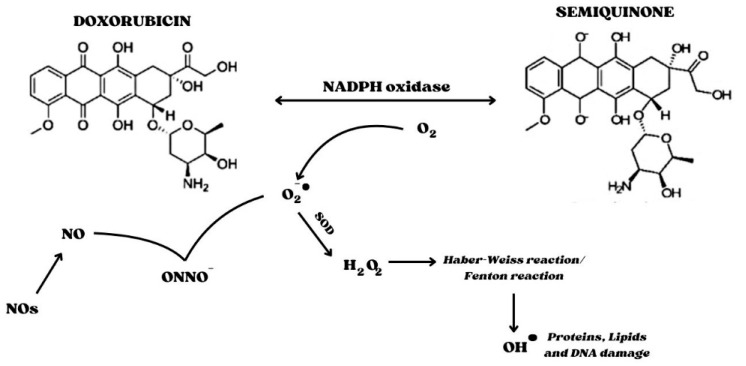
Interference of Semiquinone in the redox cycle. NADPH oxidase is involved in the generation of the Semiquinone form of Doxo. The Quinone form is regenerated by the presence of oxygen, forming superoxide, a highly reactive species. Indeed, superoxide interacts with nitric oxide to form the peroxynitrite anion, a potent free radical; in addition, superoxide is converted into peroxide by SOD. Hydrogen peroxide is responsible for the formation of hydroxyl radicals, leading to DNA, protein, and lipid damage. Abbreviations: SOD, Superoxide dismutase; NOs, nitric oxide synthetase.

**Figure 2 ijms-25-07477-f002:**
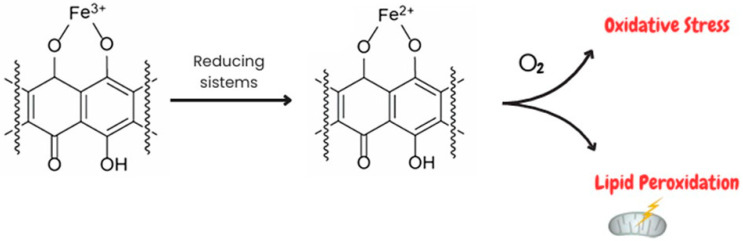
Involvement of Doxo–Fe complexes in oxidative stress and lipid peroxidation. Doxo has a high affinity for iron, leading to the formation of Doxo–Fe complexes. Doxo–Fe^3+^ complexes can be reduced by several reducing systems, such as Gpx4, to form the Doxo–Fe^2+^ complex. The latter is involved in both increasing oxidative stress through interactions with O_2_ and inducing lipid peroxidation through interactions with the negative charge of the membrane.

**Figure 3 ijms-25-07477-f003:**
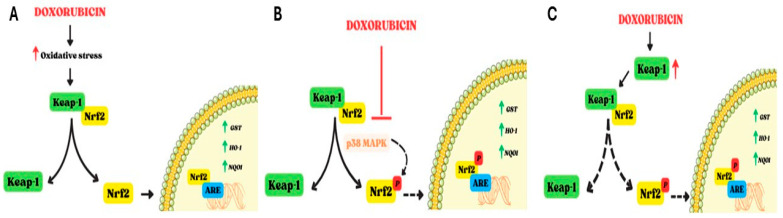
Involvement of Nrf2 in Doxo-induced cardiotoxicity. Nrf2 is negatively regulated by Keap-1; Keap-1 degradation induces activation and nuclear translocation of Nrf2, which thereby upregulates several antioxidant enzymes (GST, OH-1, and NQO1). The effects of Doxo on Nrf2 are controversial, as reported in panels (**A**–**C**). (**A**). Doxo-induced oxidative stress leads to activation of Nrf2 and its nuclear translocation to increase the levels of antioxidant enzymes. (**B**). Doxo can inhibit Nrf2 by interfering with p38MAPK. The phosphorylation of Nrf2 is reduced and interaction with ARE in the nucleus is hindered. (**C**). Doxo can induce an increase in Keap-1 levels, thereby disrupting the dissociation of the Keap-1/Nrf2 complex. In this way, nuclear translocation of Nrf2 is inhibited. Dashed lines indicate blocking of the mechanism. Abbreviations. Nrf2, Nuclear factor erythroid 2 like; Keap-1, Kelch-Like ECH associated protein 1; GST, Glutathione S-transferase; HO-1, heme oxygenase-1; NQO1, NAD(P)H quinone dehydrogenase 1; ARE, Antioxidant response elements; p38MAPK, p38 mitogen-activated protein kinase.

**Figure 4 ijms-25-07477-f004:**
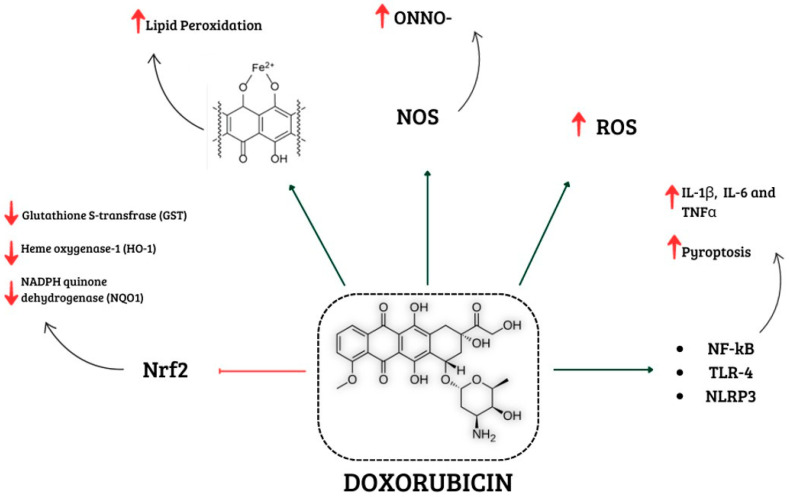
Schematic representation of different mechanisms involved in Doxo-induced cardiotoxicity. Doxorubicin is involved in increased ROS and NOS levels, which promote oxidative stress. In addition, it activates NF-kB, TLR4, and NLRP3, which are involved in enhancing levels of pro-inflammatory cytokines. Lipid peroxidation is caused by Doxo–iron complexes, and the reduction in antioxidant enzymes is related to the inactivation of Nrf2. Abbreviations: ROS, Reactive oxygen species; NOS, Reactive nitrogen species; NF-kB, Nuclear factor kappaB; TLR4, Tool-like receptor 4; NLRP3, NLR family pyrin domain containing 3; Nrf2, Nuclear factor erythroid 2 like.

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
