# Peer review of "Role of Oxidative Stress and Inflammation in Doxorubicin-Induced Cardiotoxicity: A Brief Account"

_ijms, 2024, doi:10.3390/ijms25137477_

Round 1

Reviewer 1 Report (New Reviewer)

Comments and Suggestions for Authors

The manuscript provides a comprehensive review of the mechanisms underlying doxorubicin-induced cardiotoxicity and discusses potential therapeutic strategies. Addressing the major revisions outlined above is crucial for enhancing the clarity, depth, and overall impact of the review. Once these revisions are made, the manuscript will be suitable for further consideration.

Major Revisions:

1. The section on therapeutic strategies is comprehensive but could be strengthened by including more recent clinical trial data and emerging therapeutic approaches. Summarizing these strategies in a table format, highlighting their mechanisms and efficacy, could provide a clearer comparison for readers.

2. The conclusion should emphasize future research directions more strongly. Highlighting specific gaps in current knowledge and suggesting areas for further study would enhance this section.

3. sections on oxidative and nitrosative stress, and their interplay, could benefit from more detailed mechanistic insights. For instance, further detailing how specific reactive oxygen species (ROS) and reactive nitrogen species (RNS) contribute to DIC would enhance understanding.

4. The review references various in vivo and in vitro studies but often lacks detailed descriptions of experimental designs, controls, and statistical analyzes used. Providing more comprehensive details about these studies will help readers critically evaluate the findings and their applicability.

5. While the review mentions that DIC depends on the dosage and route of Doxo administration, it would be helpful to include a more detailed analysis of how different dosages and administration methods impact the severity and onset of cardiotoxicity. This could include a summary of clinical and preclinical studies that compare these variables.

6. The review focuses on acute and chronic DIC but could include more information on the long-term cardiac outcomes for cancer survivors treated with Doxo. This could involve a discussion on long-term monitoring strategies and interventions to mitigate late-onset cardiotoxic effects.

Minor Revisions:

1. Improve the manuscript's overall English quality. Address minor grammatical issues such as "Doxo is a widely used" to "Doxo is widely used," and ensure consistency in tense and sentence structure throughout.

2. The quality of Figure 1 could be improved.

Comments on the Quality of English Language

1. Improve the manuscript's overall English quality. Address minor grammatical issues such as "Doxo is a widely used" to "Doxo is widely used," and ensure consistency in tense and sentence structure throughout.

Author Response

Replies to Referee 1

The manuscript provides a comprehensive review of the mechanisms underlying doxorubicin-induced cardiotoxicity and discusses potential therapeutic strategies. Addressing the major revisions outlined above is crucial for enhancing the clarity, depth, and overall impact of the review. Once these revisions are made, the manuscript will be suitable for further consideration.

Reply: We are grateful for your comments. We have revised the review to take account of your suggestions. Below are the changes made point by point

Major Revisions:

  1. The section on therapeutic strategies is comprehensive but could be strengthened by including more recent clinical trial data and emerging therapeutic approaches. Summarizing these strategies in a table format, highlighting their mechanisms and efficacy, could provide a clearer comparison for readers.

Reply: As suggested, the clinical trial data cited in the text have been summarised in a table (Table 1)

TRIAL

DRUG

STUDY POPULATION

RESULTS

REF

CarDHA trial

Carvedilol + DHA starting 2 and 7 days before chemotherapy

32 breast cancer patients (age 18-75 years)

Improve LVEF

[119]

Carvedilol Administration Can Prevent Doxorubicin-Induced Cardiotoxicity: A Double-Blind Randomized Trial

Carvedilol daily during chemotherapy

70 breast cancer patients

Improve LVEF

[120]

Cardioprotective Effects of Carvedilol in Inhibiting Doxorubicin-induced Cardiotoxicity

Carvedilol daily starting 24h before chemotherapy

91 breast cancer patients (age 21-69 years)

Reduce troponin I levels

[121]

Prophylactic use of carvedilol to prevent ventricular dysfunction in patients with cancer treated with doxorubicin

Carvedilol during chemotherapy

154 cancer patients

Improve LVEF

[119]

A prospective study to evaluate the efficacy and safety of vitamin E and levocarnitine prophylaxis against doxorubicin-induced cardiotoxicity in adult breast cancer patients

Vitamin E three times daily Levocarnitine four times daily before chemotherapy

74 breast cancer patients

Reduce CKMB

No significant effects on LVEF

[122]

STOP CA

Atorvastatin daily starting before Doxo

300 patients with limphoma

Improve LVEF

[126]

PREVENT

Atorvastatin daily starting 48 h before Doxo

279 cancer patients

No changes on LVEF

Modest effects on oxidative and nitrosative stress biomarkers

[127]

Table 1.  Key data from clinical trials conducted to evaluate the cardioprotective effect of drugs with antioxidant activity in patients treated with Doxorubicin

  1. The conclusion should emphasize future research directions more strongly. Highlighting specific gaps in current knowledge and suggesting areas for further study would enhance this section.

Reply: As suggested, the Conclusion has been revised.

Specifically, the sentences A better understanding of the molecular mechanisms involved in cardiotoxicity would allow to use also drugs currently in use for other therapeutic indications, such as Carvedilol or Statins, which have demonstrated interesting antioxidant and anti-inflammatory activity in several preclinical studies. Data from clinical trials confirm the prophylactic use of Carvedilol to reduce DIC, however the main gap of these results is the heterogeneity of patients enrolled and the different follow up. Results of studies using Statins also appear to be encouraging, although there are differences between data from preclinical studies and those from clinical trials, which may be due to the different molecules used. Natural compounds with antioxidant activity seems to be a promising source, but numerous studies are needed to identify compounds that can then be evaluated in humans. However, it is now clear that cardiotoxicity is also associated with other chemotherapeutic drugs, so, research in this area has a high impact in clinical practice.” have been added.

  1. sections on oxidative and nitrosative stress, and their interplay, could benefit from more detailed mechanistic insights. For instance, further detailing how specific reactive oxygen species (ROS) and reactive nitrogen species (RNS) contribute to DIC would enhance understanding.

Reply:  To better clarify the interplay between oxidative and nitrosative stress, more details have been added in the text. Moreover, the sentences “ONOO- has a high affinity for tyrosine residues in proteins and can form nitrolated proteins by nitrating tyrosine groups [69]. Reaction between ONOO- and DNA induces the production of several oxidation products of the purine and pyrimidine bases, such as 8-nitroguanine, an oxidative DNA damage’s marker [70]. Moreover, ONOO- has the potential to transform into NO2-, NO3-, and OH- with the consequent generation of active nitrogen species (RNS) ([71]). High levels of ROS/RNS induce membrane lipid peroxidation and membrane damage and trigger death cells and apoptosis [72] and may contribute to the Doxo-induced depression of cardiac function [69].”

And

“…in oxidative stress condition, the NO produced reacts with •O2, thus generating the cytotoxic free radical-peroxynitrite anion (ONOO-) that induces nytrotyrosine formation with increased mitochondrial superoxide levels in cardiac tissue [19,69,77-81].” have been added

  1. The review references various in vivo and in vitro studies but often lacks detailed descriptions of experimental designs, controls, and statistical analyzes used. Providing more comprehensive details about these studies will help readers critically evaluate the findings and their applicability.

Reply:  As suggested, the data from the preclinical studies cited in this review have been summarised in a table (Table 2).

MODEL

TREATMENT

EFFECTS

STATISTICAL ANALYSIS

REF.

Male Swiss albino mice (10 weeks age)

Allicin (10 and 20 mg/kg) once daily for 2 weeks

Doxorubicin (10mg/kg)  at 7,9 and 11th day.

AST, LDH, CK, CK-MB

IL-1β, TNF-α, 8-OHdG

COX-2, caspase-3

GSH,CAT, SOD, GPx

One-way ANOVA followed by post hoc Duncan’s test.

[130]

Male Sprangue Dawley rats (220-225g)

Acacia hydaspica (200 and 400 mg/kg) once daily for 6 weeks

Doxorubicin (3mg/kg) single dose/week

AST, LDH, CK, CK-MB

POD,CAT, SOD, QR

One-way ANOVA followed by Tukey’s test.

 [135]

Male C57BL/6J mice (22.5-23.5g)

Resveratrol (20mg/kg/day) two weeks before Doxo injection

Doxorubicin (cumulative dose of 24 mg/kg)

PTGS2, ACSL4, NCOA4

p-ERK/ERK, p-p30/p38,     pJNK/JNK

LDH

GSH, GPX4

Shapiro-Wilk test and Brown-Forsythe test. Ordinary ANOVA or Welch ANOVA test.

[129]

H9C2

Resveratrol (20µM) 6h before Doxorubicin

Doxorubicin (1µM ) 24h

p-ERK/ERK, p-38/p-38, p-JNK/JNK

LDH,PTGS2, ACSL4,   NCOA4

GSH, GPX4

Male Sprangue-Dawley rats (8 weeks)

Valsartan (20mg/kg) daily for 6 weeks

Doxorubicin (2.5 mg/kg) once per week for 6 weeks

mRNA expression levels of NOX1,NOX2, NOX4, Bax, Caspase-3, (MMP)2,MMP9, Collagen I, Beclin-1, BNP,ANP, β-MHC, GDF15, TPM1, BGN, POSTN.

ALD, TNF-α, IL-6,   BNP,ROS, MDA

   BCL2, Collagen III

   SOD

One-way or two-way ANOVA followed by Bonferroni’s pos hoc comparisons

[137]

H9C2

Valsartan (5 or 10 µM) 1h of pre-treatment

Doxorubicin(1µM)

 ROS

Male C57BL/6J mice (10-12 weeks)

Rosuvastatin (100µg/kg) 1 week before Doxorubicin and for further 2 weeks.

Doxorubicin (10mg/kg) single dose

  CAT

 NCX1, Ryr2

p-PLN

Unpaired Student t-test. ANOVA and the Tukey-Kramer multiple comparison post-test

 [124]

Sprangue Dawley rats

Dapaglifozin (0.1mg/kg) per day

Doxorubicin (3mg/kg) four weekly

p-Smad3, BNP,  α-SMA, p-38

NF-kB-p65, IL-8

One-way ANOVA followed by Bonferroni’s pos hoc comparisons

[138]

H9C2

Dapaglifoxin (0-20µM)

Doxorubicin (10µM) for 24h

ROS, pSmad3/Smad3, ANP, BNP, α-SMA, Collagen I, Fibronectin

NF-kB-p65, IL-8

p-AKT, HO-1, NQO1, SOD

C57BL/6J male mice

Resveratrol (10mg/kg)+ Fibroblast growth factor-1(0.5 mg/kg) for 7 consecutive days

Doxorubicin

(20mg/kg) single injection

CK,LDH, cTnI, cleaved caspase-3

IL-1β,IL-1α, TNF-α,Mcp1, p-IKBα, p65, ROS

BCL2/BAX ratio, CAT, SOD1,SOD2, HO-1,NQO1,Sirt1

One-way ANOVA followed by post hoc pairwise comparisons using Tukey’s test.

 [128]

H9C2

Resveratrol (20µM)

Doxorubicin (1µM)

HO-1, Nrf2 (effects was cancelled by Sirt1-shRNA

C57BL/6J male mice (8 weeks)

Rutin (100mg/kg) for 11 weeks

Doxorubicin (3mg/kg) every other day for 2 weeks starting after one week administration of Rutin

LC3 II, ATG5 , P62

Caspase-3

 Akt, Bcl-2

One-way ANOVA

[144]

C57BL/10 mice (10 weeks)

Fluvastatin (100mg/kg) 4 days before Doxorubicin application

Doxorubicin (20mg/kg) for 5 days

TNF-α

Bax

SOD2.Bcl-2

Kruskal-Wallis test in conjunction with the Mann-Whitney U post hoc test

[123]

Male Swiss mice and male Sprague-Dawley rats

Ginsenoside Rh2 (5mg/kg, 10mg/kg,20mg/kg) total 8 doses

Doxorubicin (3mg/kg or 2mg/kg) cumulative dose 18 mg/kg or 8mg/kg

AST,CK,LDH

MDA

SOD, CAT, GSH

One-way ANOVA;  Student t-test was performed

[131]

C57BL/6 mice (6-8 weeks)

Resolvin D1 (2.5µg/kg) 30 min before Doxorubicin and every day thereafter for the duration of the experiment

Doxorubicin (20mg/kg) once

LDH,CK-MB, cTnI

IL-1β,IL-6, NF-kB

MDA, NOX2, NOX4

GRP78, CHOP, caspase-12,Bax, c-caspase3

SOD,GSH, Nrf-2, OH-1,Bcl-2

One-way ANOVA followed by Tukey’s test

[136]

C57BL/6J mice (8-10 weeks)

Selenium (0.2mg/kg) 2 weeks

Doxorubicin (15mg/kg)  2 weeks

CTnI, CK, LDH

IL-1β,TNF-α, IL-18, NLP3,ASC, Caspase-1

SOD, GSH

mRNA level of Nrf-2, HO-1,NQO-1, GCLM

One-way ANOVA with Tukey’s post hoc test

[145]

Male Wistar rats (250-300g)

Panax ginseng (5g/kg) for 30 days

Doxorubicin (2.5mg/kg) for 2 weeks

MDA

GSHPx, SOD

One-way ANOVA followed by Scheffe’s multiple range test

[141]

Kunming mice

Apigenin (125 or 250 mg/kg) for 17 days

Doxorubicin (3mg/kg) cumulative dose 24mg/kg

LDH, CK,AST

Bax/Bcl-2 ratio

Beclin1, LC3B II/I

PI3K/AKT/mTOR pahway

One-way ANOVA with LSD post hoc test

 [143]

Male Sprague-Dawley rats (6-8 weeks)

Rosuvastatin (1mg/kg) for 6 weeks

Doxorubicin (1mg/kg)  for 2 weeks

AST, CK-MB

HMGB1, RAGE

TNF-α, IFN-γ

IL-10, IL-4

One-way ANOVA followed by Tukey’s post hoc test

[125]

BALB/c female mice

Ginsenoside Rh2 (20mg/kg and 30mg/kg) every day

Doxorubicin (2mg/kg) cumulative dose of 22mg/kg

Caspase-3,Capsase-7, caspase-9

IL-1β, TNF-α, IL-6

α- SMA ,  Smad2, Smad3

p 21

One-way ANOVA followed by Tukey’s multiple comparison test

[132]

H9C2

HCF

HUVEC

Ginsenoside Rh2 (2.5,5 and 10 µg/ml) 7 days after Doxorubicin treatment

Doxorubicin (100nM) for 7 days

α- SMA, Vimentin

MMP2, MMP4, MMP9, MMP14, MMP17, MMP19,MMP23, MMP27, MMP28

H9C2

Neferine (10 µM)

Doxorubicin (1µM)

NOX2, ROS, p-ERK, p-p38

Ca2+ intracellular accumulation

COX2, TNF-α, Cytochrome c, Bax

Cyclin  D1

Bcl-2            

One-way ANOVA followed by Tukey’s multiple comparison test

[133]

Table 2. Main data of preclinical studies conducted to evaluate cardioprotective effects of different molecules in Doxo-induced cardiotoxicity model.

  1. While the review mentions that DIC depends on the dosage and route of Doxo administration, it would be helpful to include a more detailed analysis of how different dosages and administration methods impact the severity and onset of cardiotoxicity. This could include a summary of clinical and preclinical studies that compare these variables.

Reply: As suggested, more details of how different dosages and administration methods impact the severity and onset of cardiotoxicity have been added. So, the sentences “Despite data on long-term effects of different Doxo dosage regimens are quite controversial because of the different follow-up times, the recommended maximum lifetime dose of Doxo is <450 mg/m2 to reduce the risk of cardiotoxic side effects [5], even if some patients exhibit morphological changes (signs of cardiac damage) with a cumulative dose of a low 200 mg/m2 [6]. An interesting systematic review summarising 11 clinical studies concluded that a Doxo infusion duration of six hours or longer reduces the risk of clinical heart failure as well as the risk of subclinical cardiac damage compared to faster infusion. Moreover, continuous infusion Doxo has been correlated to less severe endomyocardial biopsy changes than those observed in patients receiving bolus infusion [7]. Reduced risk of subclinical cardiac damage has also been correlated with Doxo peak dose ≥ 60 mg/m2 [8]. Regarding Doxo administration schedules, Von Hoff and co-workers reported that weekly schedule is associated to lowest probability for developing chronic heart failure, compared with the schedule of three times per week repeated every three week or the schedule of once every three weeks [9].” have been added.

  1. The review focuses on acute and chronic DIC but could include more information on the long-term cardiac outcomes for cancer survivors treated with Doxo. This could involve a discussion on long-term monitoring strategies and interventions to mitigate late-onset cardiotoxic effects.

Reply: As suggested, more information on the long-term cardiac outcomes for cancer survivors treated with Doxo have been added. The sentences “A study conducted in a cohort of 2000 cancer survivors highlighted that one-third of deaths were from long-term cardiotoxicity [12]. A recent review showed that 20% of cancer survivors in Europe show asymptomatic LV function reduction and this percentage is significantly higher in childhood survivors [13]. In asymptomatic patients LV wall thinning, LV diameter increase, and consequent LV wall stress increase similar to dilated cardiomyopathy, have been reported. Commons methods used for monitoring asymptomatic cardiotoxicity include serum biological biomarkers (e.g. increased cardiac troponin and natriuretic peptide), ECG, echocardiography, and cardiac magnetic resonance. Unfortunately, these methods can only confirm that a damage to the myocardium has already been done [14].  In order to prevent or limit Doxo-induced cardiotoxic effects, evaluation of cardiovascular risk and an active surveillance before, during, and following the therapeutic process is strongly recommended [15]. The early use of cardioprotective agents, such as β-blockers, renin-angiotensin-aldosterone system inhibitors (RAASi) including angiotensin converting enzyme (ACE) inhibitors and angiotensin receptor blockers (ARBs), is an established practice even if data from clinical trials are not sufficient to support clear evidence-based recommendation [16].” have been inserted in the text.

Minor Revisions:

  1. Improve the manuscript's overall English quality. Address minor grammatical issues such as "Doxo is a widely used" to "Doxo is widely used," and ensure consistency in tense and sentence structure throughout.

Reply: Manuscript has been entirely revised to avoid grammatical errors.

  1. The quality of Figure 1 could be improved.

Reply: Figure 1 has been re-edited. Hopefully, it will be clearer in this layout

Reviewer 2 Report (New Reviewer)

Comments and Suggestions for Authors

Manuscript titled "Role of oxidative stress and inflammation in Doxorubicin-induced cardiotoxicity: a brief account" is a very interesting article in the field of cardioncology. The overall structure is of good quality and easy to read. Methods and results are clear and results corroborate the initial hypothesis of the authors. Figures and tables are of sufficient quality and easy to read as well as to understand to readers.

However, manuscript needs some improvements specifically in introduction and/or discussion here the points:

1. In introduction, authors should describe the key role of myd-88 and NLRP-3 pathways in several cardiac affection related to doxorubicin alone or in combination to immunotherapy (doi 10.3389/fcvm.2022.930797). NLRP-3 and myd-88 plays a key role in pathogenesis of doxorubicin-induced cardiovascular diseases. Authors should highlight this point.

2. Authors should add the most updated cardioprotective strategies against doxorubicin-mediated cardiotoxicity like SGLT-2 inhibitors that exhibit cardioprotective properties through inhibition of anthracycline-induced ferroptosis in cancer patients and anti-inflammatory properties too (cite doi: 10.1186/s12933-021- 01346-y).

Based on these changes, the article could be suitable for publication in this journal

Author Response

Manuscript titled "Role of oxidative stress and inflammation in Doxorubicin-induced cardiotoxicity: a brief account" is a very interesting article in the field of cardioncology. The overall structure is of good quality and easy to read. Methods and results are clear and results corroborate the initial hypothesis of the authors. Figures and tables are of sufficient quality and easy to read as well as to understand to readers.

However, manuscript needs some improvements specifically in introduction and/or discussion here the points:

  1. In introduction, authors should describe the key role of myd-88 and NLRP-3 pathways in several cardiac affection related to doxorubicin alone or in combination to immunotherapy (doi 10.3389/fcvm.2022.930797). NLRP-3 and myd-88 plays a key role in pathogenesis of doxorubicin-induced cardiovascular diseases. Authors should highlight this point.

Reply: Thank you for your suggestion. Some data about the role of Myd 88 and NLRP-3 have been added to improve the part of the review dedicated to the role of inflammation in doxorubicin-induced cardiotoxicity. Great appreciation of the proposed work.

  1. Authors should add the most updated cardioprotective strategies against doxorubicin-mediated cardiotoxicity like SGLT-2 inhibitors that exhibit cardioprotective properties through inhibition of anthracycline-induced ferroptosis in cancer patients and anti-inflammatory properties too (cite doi: 10.1186/s12933-021- 01346-y).

Reply: As for the previous point, we gladly accepted the suggestion that allowed us to fill a gap. The recommended paper has been cited in the new version of the manuscript.

Based on these changes, the article could be suitable for publication in this journal

Reviewer 3 Report (New Reviewer)

Comments and Suggestions for Authors

The authors reviewed the roles of oxidative stress and inflammation in Dox-caused cardiotoxicity. The topic is important, and this manuscript will contribute to our understanding of the mechanisms. A few issues need to be addressed:

1.      Reorder the sections so that the oxidative stress discussion follows the introduction as the second part (Section 2).

2.      Expand the inflammation section (Section 2.2 or 3) to include a discussion on the causes of inflammation.

3.      The relationship between oxidative stress and inflammation needs to be discussed.

4.      Consider utilizing vector graphics for clearer figures.

Author Response

Replies to referee 2

The authors reviewed the roles of oxidative stress and inflammation in Dox-caused cardiotoxicity. The topic is important, and this manuscript will contribute to our understanding of the mechanisms. A few issues need to be addressed:

  1. Reorder the sections so that the oxidative stress discussion follows the introduction as the second part (Section 2).

Reply: Thank you for your suggestion. The review has been reorganized and is now divided as follows:

  1. Doxorubicin-induced cardiotoxicity
  2. Involvement of oxidative and nitrosative stress in Doxorubicin-induced cardiotoxicity
  3. Involvement of inflammation in Doxorubicin-induced cardiotoxicity
  4. Therapeutic strategies to counteract Doxorubicin-induced cardiotoxicity

Conclusions

Hopefully, the new version will make reading more enjoyable

  1. Expand the inflammation section (Section 2.2 or 3) to include a discussion on the causes of inflammation.

Reply: The inflammation section has been revised and more data on the causes of inflammation have been inserted

  1. The relationship between oxidative stress and inflammation needs to be discussed.

Reply: The inflammation section has been revised and more data on the relationship between oxidative stress and inflammation have been inserted

  1. Consider utilizing vector graphics for clearer figures.

Reply: We could not use vector graphics due to technical problems

We are grateful for your suggestions. We used them to improve our manuscript. We hope you enjoy the new version.

Round 2

Reviewer 1 Report (New Reviewer)

Comments and Suggestions for Authors

The authors responded to all comments, eliminated all errors and inaccuracies, and significantly improved the manuscript.

This manuscript is a resubmission of an earlier submission. The following is a list of the peer review reports and author responses from that submission.

Round 1

Reviewer 1 Report

Comments and Suggestions for Authors

The review article entitled, “Role of oxidative stress and inflammation in Doxorubicin-induced cardiotoxicity: a brief account” is having so many flaws and can’t be published in its current form.

1. The article is having highly plagiarized content.

2. It is a very very short review article missing so many important concerned areas related to the selected drug doxorubicin and only highlighting the selected and fewer topics related to oxidative stress and NRF2 only, which needs to be elaborated to other pathways involved in the same.

3. Authors can refer some latest articles on doxorubicin for better correlation. Eg.: 10.3390/cells12040659

4. Conclusion section is so small unable to conclude anything useful out of the concerned review.

Author Response

Thank you for your suggestions. We used them to improve our paper.

We hope that it is of your interest

Reviewer 2 Report

Comments and Suggestions for Authors

The manuscript of Vitale et al. describes the role of oxidative stress and inflammation in doxorubicin-induced cardiotoxicity. The paper is well-structured and results presented in the manuscript are an important contribution both for clinicians and for the scientific community interested in the development of alternative drugs to the use of conventional doxorubicin. However, there are some points that must be addressed before recommending the publication of this paper:

1)      The figures are of low quality and very unappealing. Please make them more interesting from a graphical point of view.

2)      The conclusions are very short and not very well argued. It is recommended to expand them.

3)      No references were made in the manuscript to the development of Doxil, the first FDA-approved nano-drug. This drug has been shown to be of great importance in reducing the risk related to doxorubicin-induced cardiotoxicity and was of great inspiration for scientists for the development of new doxorubicin-based formulations. In my opinion, therefore, it is necessary to devote a paragraph to Doxil, and also to scientific articles related to this drug. In this context, it is strongly recommended to cite the following articles:

-https://doi.org/10.1016/j.msec.2021.112623

-doi: 10.2174/187152012803529646.

-https://doi.org/10.1002/anbr.202100109

-doi: 10.2147/IJN.S296272

-doi: 10.1093/cvr/cvz192.

-doi: 10.1371/journal.pone.0133569

4)      Please review the caption size of Figure 4.

Kind regards.

Author Response

(The authors gave the same response as above.)

Reviewer 3 Report

Comments and Suggestions for Authors

Cardiotoxicity is a significant drawback of many chemotherapeutic drugs, particularly Doxorubicin (Doxo), a widely utilized anthracycline in cancer treatment. Despite its efficacy against tumors, the emergence of acute and chronic cardiotoxicity severely hampers its clinical utility. While prior research predominantly focused on the long-term effects of Doxo administration, recent findings emphasize that cardiomyocyte damage occurs early after a single dose, potentially progressing to heart failure. Understanding the molecular mechanisms driving early Doxo-induced cardiotoxicity is crucial for effective intervention. This review outlines various implicated mechanisms, including oxidative and nitrosative stress, inflammation, and mitochondrial dysfunction. Additionally, it highlights promising therapeutic avenues for mitigating Doxo-induced cardiotoxicity. 

The interplay between oxidative and nitrosative stress, inflammation, and mitochondrial dysfunction is intricately tied to various forms of cell death, playing a pivotal role in the progression towards heart failure. Authors could provide a more nuanced exploration by specifying and elaborating on the specific types of cell death pathways involved in the pathogenesis of Doxorubicin-induced cardiotoxicity (DIC) across the spectrum from acute to chronic phases. This detailed examination would enhance our understanding of how these cellular mechanisms contribute to the development and progression of heart failure in response to Doxorubicin exposure.

Comments on the Quality of English Language

Minor editing of English language required

Author Response

(The authors gave the same response as above.)

Round 2

Reviewer 1 Report

Comments and Suggestions for Authors

Manuscript is not yet in good shape with very less content without any concluding remarks and still having high plagiarism and can't be accepted in its current form for publication in International Journal of Molecular Sciences.

Reviewer 2 Report

Comments and Suggestions for Authors

The authors answered my questions in detail. In my opinion, with the new version, the manuscript improved in quality and readability. In particular I appreciate the introduction of a paragraph devoted to Doxil. However, as I wrote in my previous report, a lot of research has been given in recent years to technologies for the encapsulation and delivery of this drug. Therefore, according to my opinion the number of articles analyzed in this review and related to this topic is still small. Again, I invite the authors to analyse and cite at least the following articles:

-doi: 10.2174/187152012803529646

-doi: 10.2147/IJN.S296272

-doi: 10.1371/journal.pone.0133569

-doi: 10.1038/s41598-020-68017-y

I don't have any other suggestions, so once this aspect of the research is expanded, I will suggest the publication of the manuscript in the journal. Kind regards.